# Chromatin accessibility, not 5mC methylation covaries with partial dosage compensation in crows

Ana Catalán[1,2]*, Justin Merondun[2], Ulrich Knief[2,3], Jochen B. W. Wolf[1,2]

**1** Department of Evolutionary Biology, Evolutionary Biology Centre (EBC), Uppsala University, Uppsala, Sweden, **2** Division of Evolutionary Biology, LMU Munich, Planegg-Martinsried, Germany, **3** Evolutionary Biology & Ecology,Faculty of Biology, University of Freiburg, Freiburg, Germany

* ana.catalan@gmail.com

## Abstract

The evolution of genetic sex determination is often accompanied by degradation of the sex-limited chromosome. Male heterogametic systems have evolved convergent, epigenetic mechanisms restoring the resulting imbalance in gene dosage between diploid autosomes (AA) and the hemizygous sex chromosome (X). Female heterogametic systems ($AA_f Z_f$, $AA_m ZZ_m$) tend to only show partial dosage compensation ($0.5 < Z_f:AA_f < 1$) and dosage balance ($0.5 < Z_f:ZZ_m < 1$). The underlying mechanism remains largely elusive. Here, we quantified gene expression for a total of 15 male and female Eurasian crows (*Corvus (corone) spp.*) raised under common garden conditions. In addition, we characterized aspects of the regulatory epigenetic landscape quantifying chromatin accessibility (ATAC-seq) and 5mC methylation profiles. Partial dosage balance and compensation was due to female upregulation of Z-linked genes which covaried significantly with increased chromatin accessibility of the female Z chromosome. 5mC methylation was tissue and sex chromosome-specific, but unrelated to dosage. With the exception of the pseudo-autosomal region (PAR), female upregulation of gene expression was evenly spread across the Z chromosome without evidence for regional centers of epigenetic regulation, as has, for example, been suggested for the male hypermethylated region (MHM) in chicken. Our results suggest that partial dosage balance and compensation in female heterogametic systems are tightly linked to chromosome-wide, epigenetic control of the female Z chromosome mediated by differential chromatin accessibility.

## Author summary

A consequence of the degradation of one of the sex chromosomes (Y or W) is an expression imbalance between the autosomes and sex chromosomes in the heterogametic (ZW or XY) sex. Gene expression imbalance is so detrimental that mammals and fruit flies have developed complex solutions to restore expression balance. Birds on the other hand, have not developed a mechanism for complete dosage compensation but instead have developed a mechanism that results in only ~30% of Z-linked genes to be fully

---

**Data Availability Statement:** All sequencing data (RNAseq, ATAC-seq, WGBS) are available at the Sequencing Read Archive (SRA) of the National Center for Biotechnology Information (NCBI) under

Bioproject PRJNA594256 and have been mapped to the crow reference genome version 5.6 (NCBI: GCA_000738735.6). Accession numbers for individual samples are listed in S9 and S11 Tables. All proprietary code can be accessed at: https://doi.org/10.5281/zenodo.8238821. The supporting data Merged_annotation_onlyTranscripts.gtf, liver_homerAnnotation_masterbed_Str1.txt, spleen_homerAnnotation_masterbed_Str1.txt, masterbed_peakCounts.txt.gz, liver_counts_annotation_AllExpressedGenes.txt.gz, spleen_counts_annotation_AllExpressedGenes.txt.gz, has been uploaded to https://doi.org/10.5281/zenodo.8355270.

**Funding:** Funding was provided to J. B. W. W by the European Research Council (ERCStG-336536 FuncSpecGen.), the Swedish Research Council (Vetenskapsrådet; 621-2013-4510), the Knut and Alice Wallenberg Foundation (Knut och Alice Wallenbergs Stiftelse.), Tovetorp fieldstation through Stockholm University (Stockholms Universitet) and LMU Munich. The funders had no role in study design, data collection and analysis, decision to publish, or preparation of the manuscript.

**Competing interests:** The authors have declared that no competing interests exist.

compensated. In this work we found a specific increase in chromatin accessibility solely on the female's Z, where dosage compensated genes show a higher chromatin accessibility. Furthermore, 5mC methylation patterns on the Z did not show any role in the regulation of expression compensation. Thus, we uncover that chromatin accessibility covaries with dosage compensation states but not with methylation patterns, leading to the conclusion that chromatin environment has an active role in sex-specific regulation of gene expression on the Z.

## Introduction

The evolution of chromosomal sex determination commonly accompanies the degradation of one of the proto sex chromosomes resulting in male (m–XY systems) or female (f–ZW systems) heterogameity. In both systems, the heterogametic sex is left with only half of the original copy number of orthologues genes. In the absence of compensatory mechanisms, this change in gene content cuts ancestral transcript levels in half [1–3] and induces an imbalance in gene regulatory networks with genes from diploid autosomes ($X_m$:$AA_m$ = $Z_f$:$AA_f$ = 0.5) [4–6]. In the homogametic sex, the sex-chromosomal to autosomal gene dosage remains, in principle, unaffected ($XX_f$:$AA_f$ = $ZZ_m$:$AA_m$ = 1) [7].

Chromosome-level haploinsufficiency, and even dosage change in single genes, will alter stochiometric ratios of interacting proteins, generally with strong deleterious fitness effects [8,9]. A single-copy sex chromosome thus constitutes a serious peril for the hemizygous sex and is expected to prompt an evolutionary response [6,10,11]. Indeed, male heterogametic systems as divergent as fruit flies and mammals have independently evolved molecular mechanisms that function to restore expression balance [12,13]. While the degree of compensation and the underlying mechanisms differ across taxa [5,10], changes in chromatin structure and state are common themes [14–16]. For example, in mammals different types of chromatin modifications are involved in the upregulation and inactivation of the X chromosome to restore dosage compensation and balance [17–19]. In *Drosophila*, hyperactivation is limited to heterogametic males where relaxation of the chromatin structure on the X restores expression levels to the default double-dosage [12]. In *Caenorhabditis elegans* a hitherto unknown mechanism upregulates expression levels in both sexes. While this restores dosage in heterogametic males, expression overshoots in females and requires a secondary mechanism to return gene expression on the X to parity with the autosomes [20]. Throughout the text, we refer to such chromatin-based, molecular mechanisms of gene expression regulation as epigenetics in the narrow sense, not to be confounded with 'epigenetic inheritance' [21,22].

Global compensatory mechanisms restoring hemizygous to autosomal balance are not ubiquitous. In heterogametic ZW systems, we observe both, full [23–26] and partial compensation. A lack of full dosage compensation is observed across a diverse set of female heterogametic taxa including some lepidopterans [27–29], snakes [1,30] and birds [31,32]–the focal taxon of this study for which most information is available. Sex chromosomes in all birds are thought to have originated from the same pair of autosomes, at the base of avian evolution approximately 130 Ma ago [33,34]. Species show different degrees of W chromosome degradation corresponding to pseudo autosomal regions (PAR) of different sizes [35,36]. Yet, all bird species with heteromorphic sex chromosomes investigated to date show only partial dosage compensation.

Following Gu et al. [28], we differentiate between dosage balance and dosage compensation. Dosage balance refers to parity in expression between female and male genes located on the

shared sex chromosome ($X_m$:$XX_{ff}$ or $Z_f$:$ZZ_m$ = 1). Dosage compensation refers to expression parity between sex chromosomes of either sex and the ancestral, non-degenerated proto sex chromosome. In the absence of ancestral information, autosomal levels are generally used as a proxy ($X_m$:$AA_m$ = $XX_f$:$AA_{ff}$ = 1 or $Z_f$:$AA_f$ = $ZZ_m$:$AA_m$ = 1). This proxy seems justified under the conditions of equal expression for autosomal genes between the sexes ($AA_f$:$AA_m$ = 1) and lower intra-autosomal variance relative to variance between autosomes and the sex chromosome. The heterogametic bird species investigated so far tend to show both partial dosage balance ($0.5 < Z_f$:$ZZ_m < 1$) and partial dosage compensation ($0.5 < Z_f$:$AA_f < 1$). Depending on the species and tissue, f:m expression ratios of Z-linked genes range from 0.67–0.81 [31,32,37–39]. Z:A expression ratios similarly range from 0.5–0.7 in females, and reach equity in homogametic males [37–39].

The expression ratio between 0.5 (pure dosage effect) and 1 (full compensation) requires a mechanism conveying partial compensation and balance. The nature of this mechanism remains elusive, but sex-specific epigenetic variation of methylation levels is a clear candidate. As in other vertebrates, 5mC methylation in birds is largely indicative of heterochromatin and is a repressive mark that has highest functional significance within CpG islands found near gene promoters [40,41]. In the avian group Galloanserae (e.g chicken and ducks), a male hypermethylated (MHM) region has been identified on the Z and hypothesized to contribute to dosage balance [42–44]. Revision of the MHM region in chicken using chromosome-wide DNA-methylation tiling arrays, however, did not confirm a clear relationship between sex-specific methylation and regional dosage balance [45]. Moreover, a MHM region seems to be absent outside of the Galloanserae group [32,39,44].

It has further been hypothesized that partial dosage compensation in birds is most likely regulated on a gene-by-gene basis [46], possibly in combination with a chromosome-wide epigenetic mechanism (this study). The genomic tools to quantify gene expression, chromatin accessibility and 5mC methylation are readily available [40,47,48]. Yet, comprehensive studies investigating epigenetic mechanisms and chromatin structure in birds are rare [49,50] and in the context of dosage compensation near-absent [44,45]. In this study, we aim to uncover sex- and chromosome-specific differences in chromatin accessibly and methylation patterns in the Eurasian crow (*Corvus (corone) spp*.), an avian system with evidence for partial dosage balance and compensation [39]. We hypothesize that the female Z chromosome shows a more accessible chromatin environment and hypomethylation permitting female upregulation. To test this hypothesis, we quantified gene expression (RNAseq), chromatin accessibility (ATAC-seq) and 5mC methylation (bisulfite sequencing) in two somatic tissues, spleen and liver, in eight female and seven male individuals raised under common garden conditions. Synopsis of these three data types allow us to gain insight into possible epigenetic mechanisms underlying partial dosage compensation and dosage balance. We start by exploring patterns of gene expression and epigenetic features at chromosome-wide level and then zoom into patterns along the Z chromosome.

## Results

### Crow karyotype and the pseudo-autosomal region (PAR)

The crow's genome consists of 2n = 80 chromosomes including the heterotypical sex chromosomes (ZZ in males, ZW in females) and 39 pairs of autosomes [51] of which, 28 have been scaffolded to chromosome level [52]. We excluded chromosome 23 from the analysis, since most of it consists of repeats. As in other avian species and reptiles, chromosomes in the European crow differ substantially in size (4–154 Mbs) and have accordingly been somewhat arbitrarily classified into macrochromosomes (including the Z) and numerous

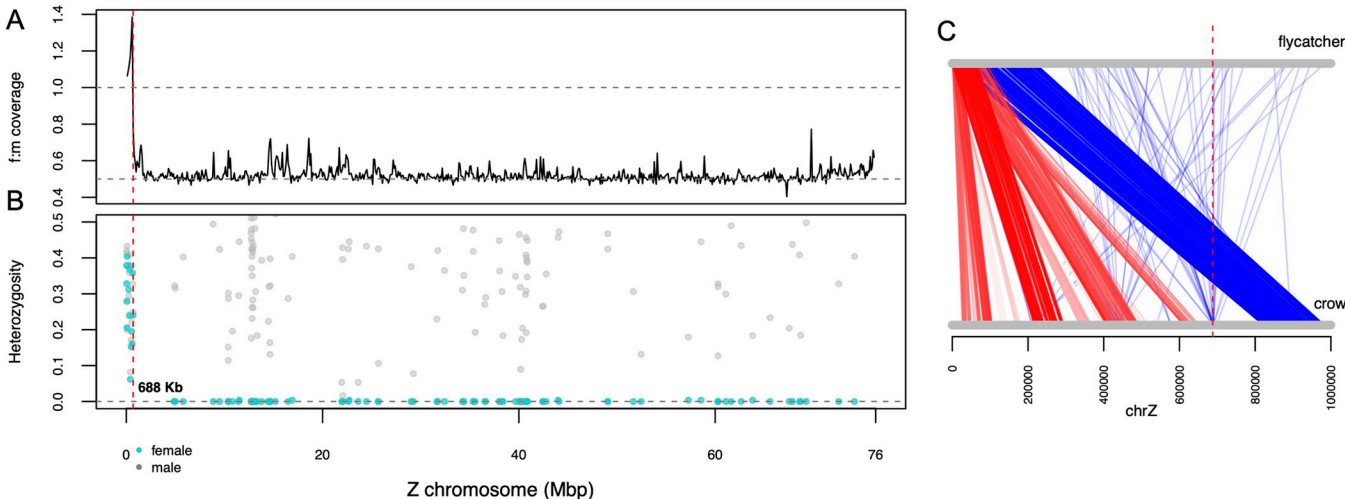

**Fig 1. Identification of the pseudo-autosomal region (PAR) on the Z chromosome of the European crow. (A)** F:m DNA sequencing coverage along the Z chromosome. **(B)** heterozygosity levels across the Z chromosome in females (light blue) and males (gray). A drop in both measures after 688 kb defines the end of the PAR. **(C)** One Mb of Z chromosome alignment between the flycatcher and the European crow. Lines between the Z represent orthologous regions found between the two species. Red: protein coding genes. Blue: non-coding regions. The vertical red dashed line marks the end of the PAR region in the European crow in all panels.

microchromosomes (<0.5μm) [53]. In the European crow, chromosomes 1–9 and the Z, belong to the macrochromosomes (25–154 Mbs), whereas chromosomes 10–28 are classified as microchromosomes (5-21Mb).

On chromosome Z, the PAR is the vestigial region that still recombines and thus reflects the remainder of the chromosomal environment before degradation of the proto-Z chromosome. It can be used as a "diploid control" to compare deviations in gene expression, chromatin accessibility and methylation patterns. We identified the crow's PAR using four independent lines of evidence based on differences in i) DNA sequence coverage (**Fig 1A**), ii) heterozygosity between females and males along the Z chromosome (**Fig 1B**), iii) homology to PAR of other passerine species (**Fig 1C**), and iv) the assembly of the crow W chromosome (for details see **S1 Text**).

## Gene expression

First, we considered gene expression levels on a chromosome-wide scale. RNA-seq data from liver and spleen overall confirm partial dosage compensation ($0.5 < Z_f:AA_f < 1$) and dosage balance ($0.5 < Z_f:ZZ_m < 1$) through female upregulation of Z-linked genes (**Figs 2A and S1, Tables 1 and S2A**). As expected, autosomes and the male copies of the Z chromosome showed no overall expression difference ($AA_m \cong ZZ_m \cong AA_f \cong 1$). Similarly, all genes expressed in the PAR had near-equal expression levels in females and males in both tissues (PAR $Z_f:ZZ_m = 1.01$ in liver, 1.01 in spleen). In females, genes located in the remaining hemizygous part of the Z chromosome also showed a pattern of (partial) reduction in expression relative to the PAR ($Z_f$ non-PAR/PAR = 0.54 in liver, 0.8 in spleen) (**S2A Table**).

To test whether partial dosage balance was due to locally upregulated gene clusters on the Z chromosome (excluding the PAR) we first categorized genes into fully balanced (147/159 genes in liver/spleen), partially balanced (99/122 genes) and unbalanced (277/350 genes) (see methods). Using a sliding window analysis employing windows of different sizes (PAR equivalents of 688 kb or 1 Mb) no significant clustering of compensated genes was found (**Fig 3A,** Fisher's exact tests with multiple testing correction **S3 Table**). The same result was obtained

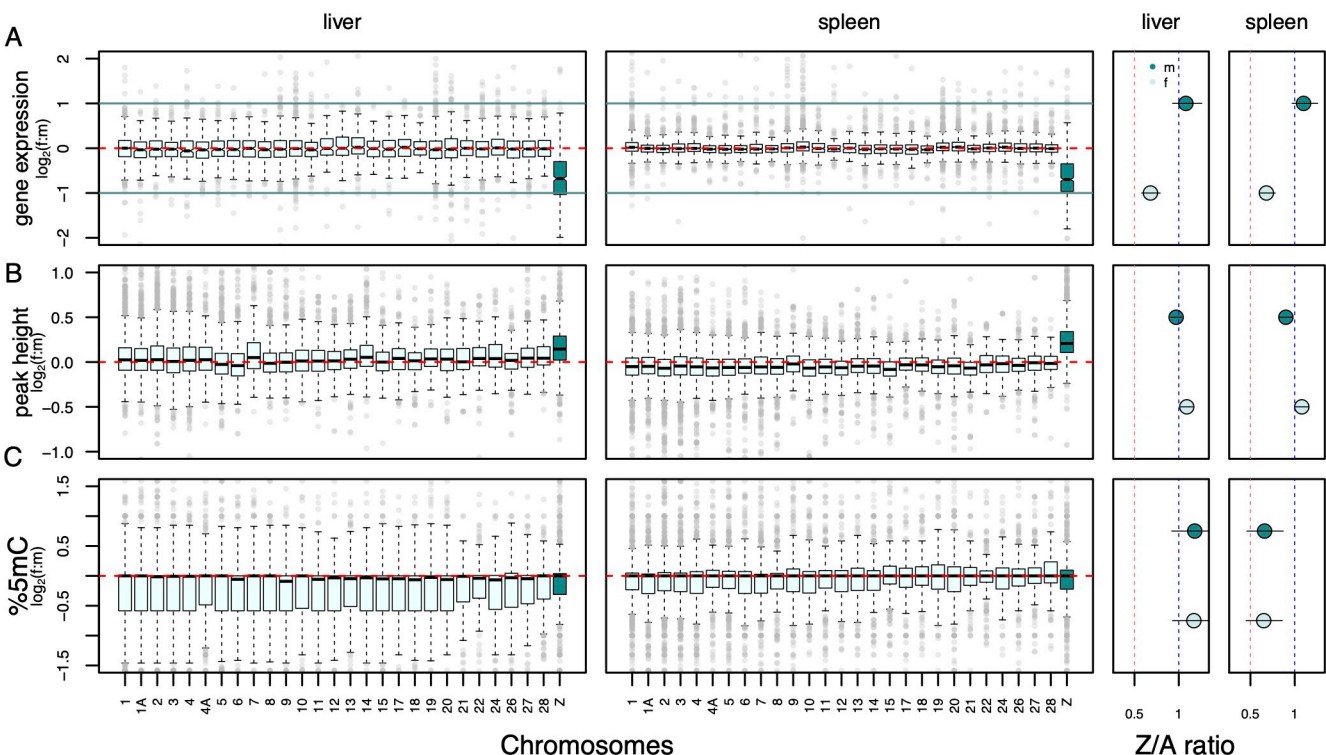

**Fig 2. Patterns of dosage effects for gene expression, chromatin accessibility (peak height) and 5mC methylation shown for 27 autosomes and the Z chromosome.** The two leftmost columns show f:m ratios of genes by chromosome for liver and spleen, respectively, with the Z chromosome highlighted in color. Horizontal red dashed lines indicate an equal f:m ratio. Boxes encompass the interquartile range across genes, whiskers extend to 1.5 x the interquartile range. The two rightmost columns show Z-chromosomal to autosomal (Z:A) model estimate ratios separately for females ($Z_f:AA_f$, light shading) and males ($ZZ_m:AA_m$, dark shading). Here, red vertical lines mark a ratio of Z:A = 0.5 and blue vertical lines mark a ratio of Z:A = 1. Bars in dot plots represent 95% confidence intervals of parameter estimates of statistical models controlling for co-variates drawn from 10,000 bootstraps (**S2A–S2C Table**). (A) Log$_2$(f:m) gene expression, (B) log$_2$(f:m) ATAC-seq peak height in the regulatory region of a gene and (C) f:m 5mC methylation percentages in CpG islands. Consult Table 1 for model-based estimates and confidence intervals for $AA_f:AA_m$, $ZZ_m:AA_m$, $Z_f:AA_f$ and $Z_f:ZZ_m$ ratios. Additional results from statistical analyses including other regions of the gene can be found in S2 Table.

**Table 1. Estimates of chromosome and sex-specific ratios of gene expression, ATAC-seq peak height and 5mC methylation levels in liver and spleen tissue.** Values are shown for the regulatory region of a gene (peak height: transcription start/end sites, 5mC methylation: CpG islands); for values integrating across the entire gene consult **S2 Table**. The color code categorizes 95% confidence intervals (shown in brackets) into biologically meaningful categories consistent with ratios of 1 (blue),] 0.5;1] (yellow) or larger than 1 (green). Autosomal ratios $AA_f:AA_m$ and $ZZ_m:AA_m$ constitute the null expectation and should not differ from unity for any of the three measures (blue). For gene expression, values within] 0.5;1] (yellow) indicate partial dosage balance and compensation. For peak height values above 1 (green) indicate a more permissive chromatin environment on the female Z chromosome with the potential to induce partial dosage balance and compensation. For 5mC methylation, values < 1 (yellow) indicate tissue-specific hypomethylation on the Z chromosome in both sexes. A: autosomes, Z: Z chromosome, f: female, m: male.

| Trait | Tissue | | | dosage balance | dosage compensation |
|---|---|---|---|---|---|
| | | **AAf:AAm** | **ZZm:AAm** | **Zf:ZZm** | **Zf:AAf** |
| **Gene expression** | liver | 1.01 (0.99–1.03) | 1.08 (0.94–1.26) | 0.63 (0.62–0.65) | 0.68 (0.59–0.79) |
| | spleen | 1.00 (0.98–1.02) | 1.10 (0.94–1.28) | 0.62 (0.61–0.64) | 0.69 (0.61–0.64) |
| **peak height** | liver | 0.96 (0.86–1.08) | 1.03 (0.94–1.12) | 1.04 (0.93–1.18) | 1.12 (1.02–1.21) |
| | spleen | 0.94 (0.86–1.03) | 1.09 (0.96–1.23) | 1.06 (0.97–1.15) | 1.22 (1.08–1.33) |
| **5mC methylation** | liver | 0.94 (0.82–1.07) | 1.18 (0.92–1.42) | 0.93 (0.81–1.07) | 1.17 (0.93–1.43) |
| | spleen | 0.95 (0.66–1.38) | 0.66 (0.46–0.87)* | 0.94 (0.58–1.53) | 0.65 (0.45–0.86)* |

*hypomethylation in spleen is also observed in males ($ZZ_m:AA_f$) precluding a role in dosage compensation

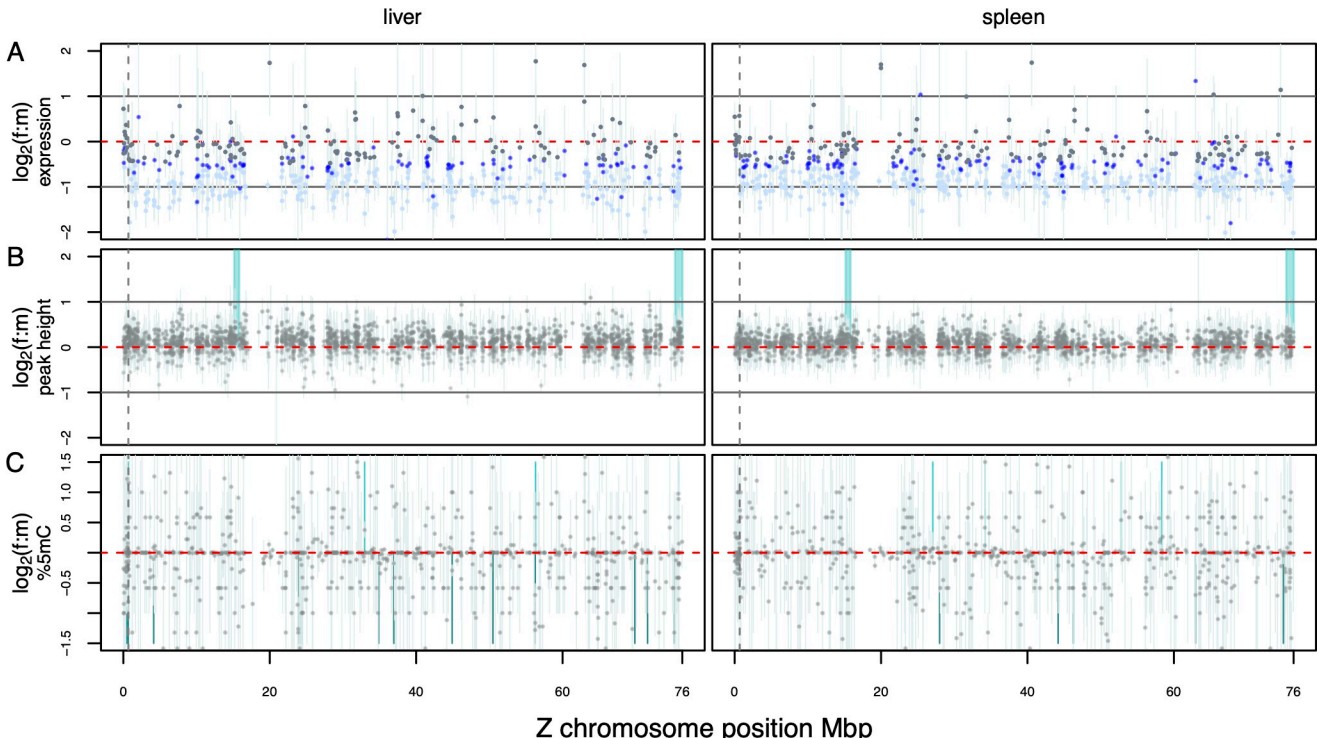

**Fig 3. Mean log₂(f:m) along the Z chromosome for all three axes of consideration.** The dashed, vertical line separates the PAR region (left) from the non-PAR, hemizygous region (right). (**A**) Gene expression. Gray dots represent dosage balanced genes, royal blue dots genes with partial balance and light blue dots unbalanced genes. (**B**) ATAC-seq peak height at regulatory regions. Vertical blocks mark female-biased regions. (**C**) 5mC methylation of CpG islands. **A–C**: For each estimate, 95% confidence intervals were calculated as the median of bootstrapped female and male values (10,000 times) of gene-centered estimates. Shaded vertical areas represent regions shared between organs with significant enrichment of compensated genes in males (dark blue, values < 0) or females (light blue, values > 0).

when the enrichment analysis was done using windows containing 15 expressed genes (mimicking the number of expressed genes on the PAR; **S3 Table**). A lack of regional concentration of compensated genes was further corroborated by a lack of autocorrelation in $Z_f$:$ZZ_m$ values (**S2 Fig**).

From all dosage-balanced genes present in both tissues, 30% (194 genes) were shared between tissues and showed significant gene ontology enrichment (GO) involving processes related to cell skeleton and organelle organization (**S4 Table**). We recovered a negative correlation between dosage balance (f:m ratios) with gene expression level measured as FPKM (linear model, liver: p-value = 8.508e-09 / cor = -0.15, spleen: p-value = 1.267e-07 / cor = -0.13).

Overall, these results confirm a pattern of partial dosage balance that is randomly distributed along the Z chromosome, without showing regions or islands of enrichment.

## Chromatin accessibility

We hypothesized that the observed upregulation on the female's Z chromosome gene expression might be under epigenetic control. Specifically, we first tested whether it may be associated with a more permissive chromatin environment on the female Z chromosome. We therefore performed an ATAC-seq experiment providing information about chromatin accessibility along the genome [54]. Details of the pre-processing and QC steps taken for the ATAC-seq data, are presented in **S2 Text**. We used the amplitude of ATAC-seq signals (peak height) as a measure to assess the local chromatin environment. Accordingly, chromatin

accessibility was measured as peak height of orthologous peaks considering two levels: (1) gene: peaks spanning the gene body and region 20kb up- and downstream of the expressed gene (2) regulatory: peaks residing only within 1 kb upstream of transcription start sites (TSS) or 1kb downstream of transcription termination sites (TTS).

On average, we observed 7 linked peaks per expressed gene (**S3 Fig**). The average height of ATAC-seq peaks identified across autosomes did not differ between the sexes. Peak height in females, however, was significantly elevated in Z-linked genes over autosomal genes ($Z_f:AA_f>1$) and Z-linked orthologues in males ($Z_f:ZZ_m>1$) (**Tables 1 and S2B, Figs 1B and S4**). This effect of elevated peak height on the female Z was not due to few outlier regions with extreme values of peak height, but was persistent across all quantiles of the peak height distribution (**S5 Fig**). Higher peaks were also stable across varying intervals around genes including 1–15kb up- and downstream of the gene model (**S6 Fig**). Higher accessibility in females was also observed at the level of ATAC-seq read coverage excluding possible artifacts of the peak calling algorithm. Female:male coverage ratio was centered around 1 for autosomes, but was elevated on the female Z beyond the expectation from chromosomal copy number of 0.5 (**S7 Fig**).

Similarly to gene expression and to patterns observed on the autosomes, peak height did not differ between the sexes in the PAR, nor in the non-PAR region in males $ZZ_m:PARZZ_m$ ~1). Nevertheless, in females the non-PAR region was significantly elevated over ploidy expectations ($Zf:PARZf >0.5$) (**S2B Table**). Analogous to the pattern of gene expression along the Z chromosome, the open chromatin environment showed no clear clustering. Using a sliding window approach (window size 688 kb, shift 344 kb), we uncovered two regions with significantly higher peaks for females in both liver and spleen (light blue areas in **Fig 3B and S5 Table**). Of all 1,761/1,770 peaks shared between females and males, 90/78 peaks in liver/spleen showed a female-biased tendency and only 16/17 showed male-bias (Wilcoxon test, non-significant after multiple testing (**S6 Table**).

Hypothesizing a functional relationship between chromatin accessibility and gene regulation, we expected peak height to be positively correlated with gene expression. Gene expression covaried with peak height (**Fig 4A**) and the highest peak representing the region with the most accessible chromatin was more often found in promoter regions than other parts of the gene or intergenic regions (**Fig 4B**).

To further assess the functional relationship between the ATAC-seq assays and gene expression, we conducted a differential accessibility analysis of the ATAC-seq data, using raw read counts in peaks, as an additional measure of chromatin accessibility in a set of orthologous

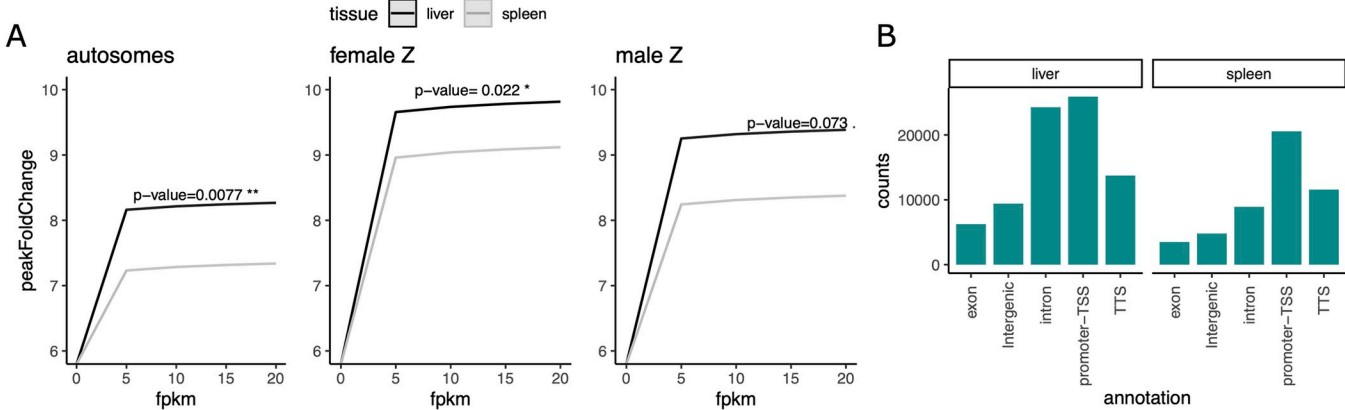

**Fig 4.** (**A**) Predicted values for peak fold change explained as a function of gene expression. Data is shown for the first two expression quantiles. (**B**) Barplots separating the highest peak of any expressed gene by functional category. TSS: transcription start site. TTS: transcription termination site.

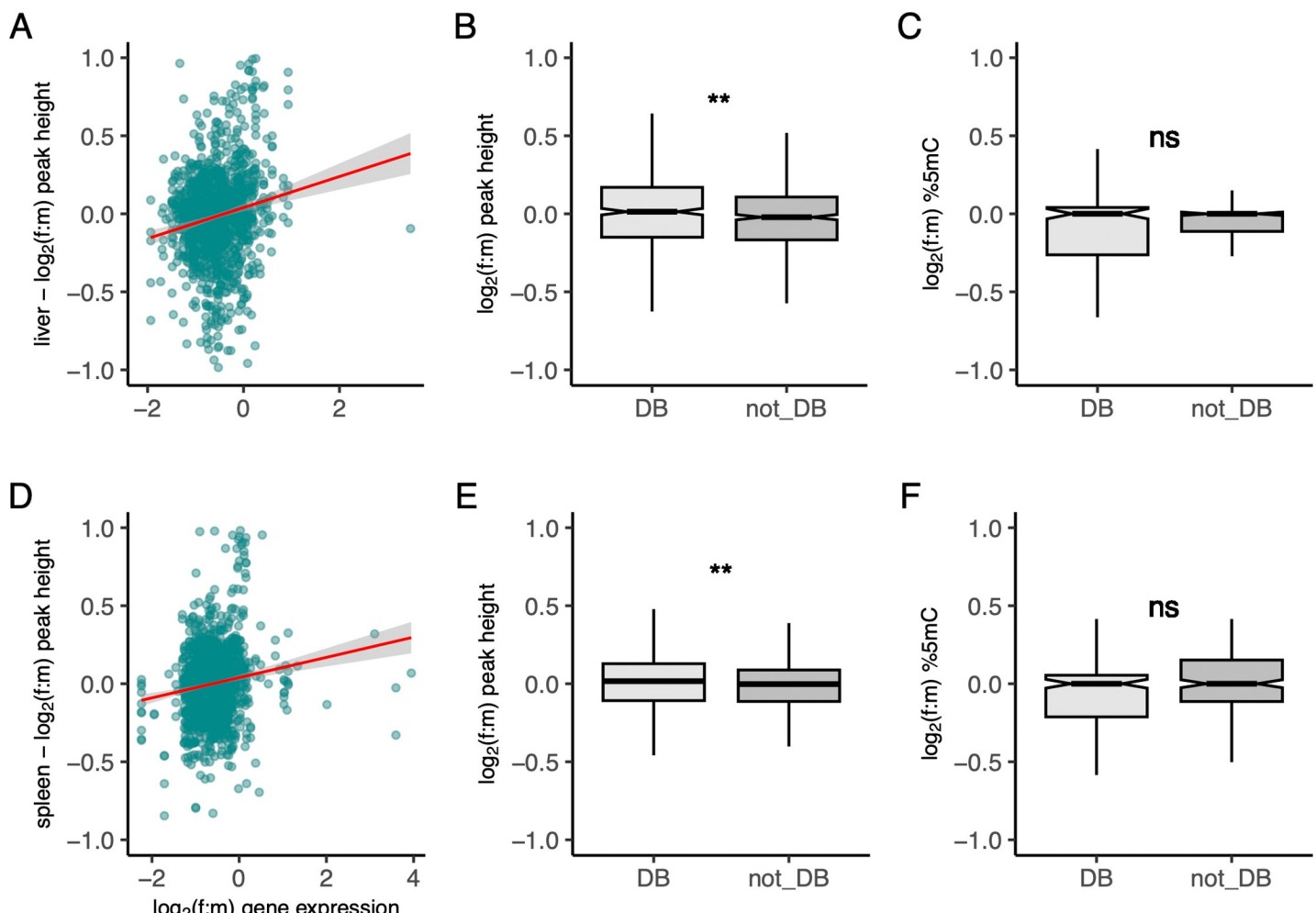

**Fig 5. Relationship between epigenetic features, gene expression and dosage balance (DB) on the Z chromosome.** (**A,D**) Linear regression of $\log_2$(f:m) peak height to gene expression. Peak height is measured as read counts in peaks. (**B,E**) Boxplots depicting $\log_2$(f:m) ratios of peak height in regulatory regions of dosage balance (DB) and not DB genes. (**C,F**) $\log_2$(f:m) ratios of 5mC methylation in CpG islands of DB and not DB genes. Boxes in boxplots encompass interquartile ranges across genes, whiskers extend to 1.5 x the interquartile range. Wilcoxon-test statistical significance levels: 0.0001 '\*\*\*', 0.001 '\*\*', 0.01 '\*'. Data show for liver (**A**, **B**, **C**) and spleen (**D**, **E**, **F**).

peaks (**S7 Table**). In dosage-balanced genes, females over male ratios were significantly higher counts than in non-balanced genes corroborating the results from above (**Fig 5B and 5E**, Wilcoxon-test, P-value: $2.7 \times 10^{-3}/6.6 \times 10^{-3}$). Also, when fitting a linear regression, we recovered a significantly positive correlation between f:m log fold-change and f:m gene expression (linear model, P-value: $1.2 \times 10^{-14}/1.3 \times 10^{-15}$ $r^2_{liv} = 0.046$, $r^2_{spl} = 0.037$), possibly suggesting higher balance for more expressed genes (**Fig 5A and 5D**).

Overall, these results are consistent with the hypothesis of a more accessible chromatin environment along the hemizygous part of the female Z chromosome which influences gene expression levels without any pronounced clustering.

## 5mC methylation

DNA methylation constitutes another candidate regulatory mechanism underlying partial dosage compensation. To assess its role, we generated whole-genome bisulfite sequencing data

in liver and spleen for two female and two male individuals. We assayed the degree of 5mC methylation at the gene level, and separately for CpG islands which are enriched in promoter regions [55]. As expected [56], gene methylation levels were high (0.66 liver, 0.65 spleen), much reduced (and bimodal) in regulatory CpG islands (0.20 liver, 0.22 spleen) (**S8 Fig**) and near-zero within ATAC-seq peaks (**S9 Fig**).

At the gene level, autosomes and sex chromosomes showed equal methylation levels across both sexes and tissues (**S2C Table**). In CpG islands, methylation patterns showed strong tissue specificity. In liver, methylation was similar across chromosomes and sexes, whereas in spleen Z-linked CpG islands were significantly hypomethylated compared to autosomes in both sexes (**Fig 2C, Tables 1 and S2C**). Within the context of dosage balance, we observed no significant difference in dosage balanced Z-linked genes compared to unbalanced genes (**Fig 5C and 5F** (minimum p-value: 0.428) and averages hardly differed (mean f:m: -0.01%) (**S8 Table**). Furthermore, we identified no correlations between sex-based differences in methylation and gene expression across either dosage balanced or unbalanced Z-linked transcripts (absolute value Spearman's rho range: 0.0086–0.13, median: 0.051) (**S10 Fig**).

We also found no evidence for a local, sex-specific clustering of 5mC methylation comparable to the male hypermethylated regions (MHM) in chicken, which in fact appears to be a female hypomethylated region of the Z chromosome with the potential to establish dosage balance [44,57]. Using blast, we found no hits corresponding to the chicken MHM region in the crow genome. Moreover, in accordance with the gene expression and ATAC-seq analyses, we found no significant regional enrichment of male hypermethylation or female hypomethylation (**Fig 3C**). These results were further corroborated with traditional differential methylation analyses. Using a beta-binomial regression on read counts we identified several localized regions of DNA methylation that differed between the sexes along the autosomes and Z chromosome. However, both male and female methylation patterns on the Z were well within the variation of autosomes and showed no clustering indicative of a localized organizational center resulting in dosage balance (**S11 and S12 Figs**).

Overall, these results indicate tissue- and sex chromosome-specific regulation of 5mC methylation, yet no evidence for a role of 5mC methylation contributing to dosage compensation or dosage balance. There was also no evidence for regional sex-specific organization arising from differences in methylation.

## Discussion

In this study, we quantified chromosome and sex-specific variation in gene expression (RNA-seq), and used two methods to investigate the underlying epigenetic variation including chromatin accessibility (ATAC-seq) and levels of 5mC methylation (bisulfite sequencing). We interpret our findings in light of dosage compensation and dosage balance.

### Gene expression: $0.5 < Z_f$: $(ZZ_m = AA_{f = m}) < 1$

Gene expression data provided evidence for incomplete dosage compensation ($0.5 < Z_f$:$AA_f < 1$) and dosage balance ($0.5 < Z_f$:$ZZ_m < 1$) on a background of autosomal parity between the sexes ($AA_f = AA_m$). The expression pattern of $Z_f < (ZZ_m = AA_{f = m})$ has been observed in many ZW chromosome systems such as birds, snakes and *Heliconius* butterflies, irrespective of the age of the non-degraded sex-chromosome [1,27,32]. Deviations from this pattern have been suggested for moths [24,58], livebearer fish [59] or the brine shrimp [23]. Our statistical analyses controlling for confounding genomic features showed that gene expression on the single-copy part of the female Z chromosome was significantly increased over the expectation of gene dose alone. This was observed relative to the expression in males for genes residing on

the Z ($0.5 <$ Z$_f$:ZZ$_m < 1$, partial dosage balance), relative to overall expression of autosomal genes when compared to the Z ($0.5 <$ Z$_f$:AA$_f < 1$) and relative to genes on the dual-copy, pseudo-autosomal region of the female Z chromosome ($0.5 <$ non-PAR$_f$:PAR$_f < 1$, partial dosage compensation). Using autosomal levels of gene expression as an approximation for ancestral, diploid expression levels, this result argues for upregulation of genes located on the hemizygous Z chromosome in females during the course of sex chromosome evolution. This conclusion is consistent with previous point estimates from other avian species exceeding Z$_f$:AA$_f$ expression ratios of 0.5 for the majority of genes [32,37–39,60].

## Candidate epigenetic mechanisms

Variation in chromatin accessibility and DNA methylation patterns is an important determinant of variation in gene expression [61], and thus constitutes a plausible mechanism to equalize differences in gene dose [12,14,16,17]. Here, we quantified broad-scale patterns of chromatin accessibility and 5mC methylation serving as proxies for epigenetic mechanisms with putative direct and indirect effects on transcription [15,40].

A challenge that we addressed while interpreting patterns of epigenetic variation across sexes and chromosomes is the influence of confounding variables. Unsurprisingly, variation in both ATAC-seq and 5mC methylation data was in part explained by chromosome and gene length (**S2 Table**). These variables are closely related to recombination rate and are known to influence many aspects of genome biology in birds [62,63]. For example, in the neighborhood of genes, peak height was influenced by chromosome length and gene length. In short, open chromatin was sparser, but more accessible the longer the genes and chromosomes. The degree of 5mC methylation also depended on chromosome length with longer chromosomes showing a tendency to harbor more hypomethylated genes. Clearly, these confounding variables need to be statistically accounted for to obtain sex-specific parameter estimates isolating dose effects between the Z chromosome and the highly variable autosomes. The use of flexible, mixed-effects linear models served this purpose and has been advocated in the context of dosage compensation before, though has rarely been put into practice [10].

## Chromatin accessibility

The approach of relating measures of epigenetic variation to dosage compensation and balance is based on the rationale that these two measures are, in essence, functionally related to gene expression. Several lines of evidence supported this hypothesis, but also highlight limitations of the empirical data. ATAC-seq peak height is known to translate into higher chromatin accessibility [64], and other studies have seen a correlation between the number of chromatin accessible sites and the level of gene expression [65,66]. In this study, we found a positive correlation between gene expression and chromatin accessibility, both when measured as local fold enrichment to the background and measured as raw read counts mapped within peaks. We also show that dosage compensated genes have higher levels of open chromatin in comparison to non-compensated genes.

Along the Z chromosome, we found no systematic clustering of dosage compensation nor of chromatin accessibility states. We did, however, uncover two regions with significantly elevated peak height values in both tissues. These may point towards female-bias localized regions, which could function as additional coordination centers for gene regulation. Overall, these findings propose a dosage compensation and balance mechanism that is spread throughout the entire Z chromosome with the putative involvement of localized 'organization centers' for dosage balance. This observation is compatible with a targeted gene-by-gene mechanism as previously been suggested for birds [5,46,67], or an overall more accessible chromosome-wide

environment affecting upregulation of a subset of genes. Since increased chromatin accessibility was enriched, but not restricted to regulatory regions of female Z-linked genes, we lean towards a model of chromosome-wide regulation that is further modified on a gene by gene basis. Comparative evolutionary analyses between species are needed to disentangle the order of events. Our observations are, however, incompatible with different levels of compensation strength corresponding to evolutionary strata on the Z chromosome [57,68].

## 5mC methylation: no global nor regional dosage effect

Local variation in 5mC methylation is of undoubted importance for gene expression [69], and has previously been discussed in the context of dosage compensation [70]. ATAC-seq peaks were constitutively demethylated (**S9 Fig**) while their flanking sequences mirrored genome-wide averages, further strengthening the functional relationship to gene expression [44]. Clearly, in the absence of a detailed regulatory map in crows, inferring process from (broad-scale) patterns is very challenging, and results need to be interpreted with caution. Many of the open chromatin and methylated regions will not bear any direct functional relationship to gene expression, as has been shown by other studies [15]. To mitigate potential biases and to reduce background noise concealing functionally active genomic regions, we quantified patterns at functional resolution from genes to regulatory regions (CpG islands).

As expected, methylation differed strongly among levels of integration. Gene bodies were most strongly methylated, followed by CpG islands and by hypomethylated regions of accessible chromatin. Chromatin accessibility is expected to show a strong negative relationship with DNA methylation [71], as DNA methylation hinders the anchoring of the transcription machinery and tightens up chromatin structure [14,72]. Despite these expected genome-wide correlations, patterns of DNA methylation levels were neither sex-specific, nor did they show any relationship to the pattern of partial dosage compensation observed for gene expression and ATAC-seq data. Whole-genome methylation data rather portrayed a conserved methylomic landscape on the Z chromosome between sexes. This is consistent with previous reports in chicken and the white-throated sparrow [44]. Yet, we found no evidence for male-specific hypermethylation on the Z chromosome, as suggested for chicken [42–44] and other species in the Galloanserae clade [57]. Instead, we identified regions of increased differences in methylation consisting of hypo- and hypermethylated sites in both sexes. The lack of an MHM region further indicates that in the European crow the DNA methylation landscape on the Z chromosome does not contribute to regional sex-specific gene expression to achieve dosage compensation. This is consistent with mechanisms detected in the zebra finch, where methylation is autonomous from dosage compensating mechanisms in brain tissue [73]. Instead, we hypothesize that the female's Z has evolved a chromatin environment that increases permissiveness along the entire chromosome and that gene-specific control of expression is fine-tuned by specific chromatin accessible regions. The fact that 5mC methylation levels co-vary with open-chromatin features across sexes and chromosomes would, however, allow for a possible indirect contribution.

## Conclusions

This study shows epigenetic differences in the Z chromosome between females and males of a ZW chromosome system with increased chromatin accessibility in the female Z. We propose that higher accessibility on the female Z might procure an environment for the upregulation of specific genes as we recovered a positive correlation between higher chromatin accessibility and gene expression. 5mC methylation was for most part equalized in females and males, leading to the hypothesis that 5mC methylation does not play a direct and specific role in dosage

balance or compensation. Much rather, we observed strong tissue-specific hypomethylation of the sex chromosome methylation in both sexes alike. We also did not recover regional 'centers' of regulation on the Z chromosome, such as the male hypermethylated region observed in chicken. The data also provided no hint at an influence of discrete genomic regions differing in the age of recombination cessation (evolutionary strata [74]). In summary, this work highlights the role of chromatin dynamics in the process of partial dosage balance and compensation and constitutes a first step towards a functional understanding. Future work may also consider additional axes of epigenetic variation, such as histone modification with known effects on transcription [75]. Such assays, where feasible, could further help to pin down the functional elements conveying partial dosage compensation and dosage balance that is broadly observed in ZW systems. Profiling genome wide chromatin accessibility and methylation patterns in different organs will further contribute to our understanding of the functional mechanisms involved in bird partial dosage compensation and pinpoint common patterns. The analysis of proteomes will also add another tier of knowledge on the role of translational regulation in dosage compensation and balance [76–78].

## Material and methods

### Ethics statement

*Regierungspräsidium Freiburg* granted permission for the sampling of wild carrion crows in Germany (Aktenzeichen 55–8852.15/05). Import into Sweden was registered with the *Veterinäramt Konstanz* (Bescheinigungsnummer INTRA.DE.2014.0047502) and its Swedish counterpart *Jordbruksverket* (Diarienummer 6.6.18-3037/14). Sampling permission in Sweden was granted by *Naturvårdsverket* (Dnr: NV-03432-14) and *Jordbruksverket* (Diarienummer 27–14). Animal husbandry and experimentation was authorized by *Jordbruksverket* (Diarienummer 5.2.18-3065/13, Diarienummer 27–14) and ethically approved under the Directive 2010/63/EU on the Protection of Animals used for Scientific Purposes by the *European Research Council* (ERCStG-336536).

### Taxonomic considerations

The two taxa considered here–carrion crows (*Corvus (corone) corone*) and hooded crows (*Corvus (corone) cornix*)–have been considered separate species by some [79,80]. Genome-wide evidence, however, rather supports treatment as a single species with two segregating colour morphs [81–83]. Relevant for the context of this study, previous work has also found near-identical gene expression profiles [84] with marked differences only for genes expressed in melanocytes underlying plumage divergence [85,86]. For the purpose of this study, we therefore consider carrion and hooded crows as members of the same species.

### Data set and sample collection

In May 2014, crow hatchlings of an approximate age of 21 days were obtained directly from the nest [87]. Hooded crows (*C. (corone) cornix*) were sampled in the area around Uppsala, Sweden (59˚52'N, 17˚38'E), and carrion crows (*C. (corone) corone*) in the area around Konstanz, Germany (47˚45N', 9˚10'E). A single individual was selected from each nest to avoid any confounding effects of relatedness. After transfer of carrion crows to Sweden by airplane, all crows were hand-raised indoors at Tovetorp field station, Sweden (58˚56'55"N, 17˚8'49"E). When starting to feed by themselves, they were released to large roofed outdoor enclosures (6.5 x 4.8 x 3.5 m), specifically constructed for the purpose. All crows were maintained under common garden conditions in groups of a maximum of six individuals separated by sub-

species and sex. In October 2016, at an age of approximately 2.5 years, individuals were euthanized by cervical dislocation. Tissues were dissected from eight females (3 *corone*, 5 *cornix*) and seven males (4 *corone*, 3 *cornix*). Tissue for RNA extraction was conserved in RNAlater (ThermoFisherScientific) and stored at -80˚C. Tissue designated for ATAC-seq was flash frozen at -80˚C immediately after dissection. Accession numbers for the following sequencing experiments (RNA-seq, ATAC-seq, WGBS) can be retrieved in **S9 Table**.

## RNA-seq

**Data generation and processing.**   RNA-seq reads were generated for liver and spleen. Frozen tissue was disrupted with a TissueRuptor (Qiagen, Hilden, Germany), and total RNA was extracted with the Rneasy Plus Universal Kit (Qiagen, Hilden, Germany). RNA quantity and quality were determined with an Agilent Bioanalyzer 2100 (Agilent Technologies, Santa Clara, CA). RNA-seq libraries were prepared from 500 ng total RNA using the TruSeq stranded mRNA library preparation kit (Illumina Inc, Cat# 20020594/5) including polyA selection. The library preparation was performed according to the manufacturers' protocol (#1000000040498). Fifty base pair single-end reads were generated on a HiSeq2500, v4 sequencer (Illumina). Sequencing was performed by the SNP&SEQ Technology Platform in Uppsala, Sweden.

Base pair quality was assessed with FastQC v0.11.5 [88] and bases with a Phred quality score < 20 were removed. All reads with a minimum length of 20 bp were kept for mapping. Reads were mapped to the crow genome version 5.6 (NCBI: GCA_000738735.6) with STAR [89]. Version 5.6 was built from the previous version 5.5 (NCBI: GCA_000738735.5, [52] with an additional round of Illumina read-polishing using Pilon v1.22 [90]. An annotation lift-over of gene models from crow genome version 2.5 [82] to the version 5.6. was performed. We used StringTie [91] to predict novel isoforms, that were merged into the annotation lift-over file if not overlapping with already annotated features.

Raw read counts per annotated transcript were calculated with HTSeq (v0.9.1) [92]. Fragments Per Kilobase per Million mapped reads (FPKM) were calculated and normalized with Cufflinks [93]. For empirical summary statistics of central tendency, we filtered out genes if the mean FPKM of the focal gene across all females and across all males was <1. Thus, genes with FPKM > 1 were considered expressed. For all model-based inference, "actively expressed genes" were identified by calculating zFPKM-values for each sample and tissue using the *zFPKM* R package (v1.10.0) and we considered genes with zFPKM > -3 as expressed [94].

**Quantification of dosage compensation and dosage balance.**   For each tissue, f:m expression ratios were calculated per chromosome (A1A1$_f$:A1A1$_m$, A2A2$_f$:A2A2$_m$, . . . Z$_f$:ZZ$_m$) as $\log_2$(median(Gene$_i$FPKM$_{female}$)/median(Gene$_i$FPKM$_{male}$)) for each gene. Sex-specific Z:A ratios (Z$_f$:AA$_f$, ZZ$_m$:AA$_m$) were calculated as median($\log_2$(FPKM of all Z-linked genes)) / median($\log_2$(FPKM of all A$_i$-linked genes)). These raw data are influenced by chromosome features which have been accounted for by statistical models (see below). Throughout the manuscript model-based estimates are reported. Model based estimates, as well as summary statistics of the raw data can be found in **S2 Table**. The degree of dosage compensation was categorized by partitioning $\log_2$(f:m) expression of each gene into three categories: (1) fully balanced: $-0.25 < \log_2$(f:m) $< 0.25$, (2) partially balanced: $-0.5 < \log_2$(f:m) $< -0.25$ and $0.25 > \log_2$(f:m) $> 0.5$ and (3) unbalanced: $\log_2$(f:m) $< -0.75$ and $\log_2$(f:m) $> 0.75$. All $\log_2$(f:m) $> |2|$ were classified as female or male biased genes.

An analysis of gene ontology enrichment (GO) was performed for dosage balance on the following categories: (1) all dosage-balanced genes in liver OR spleen (2) dosage-balanced genes in liver AND spleen, (3) all dosage-unbalanced genes in liver OR spleen (4) dosage-

unbalanced genes in liver AND spleen. GO enrichment analysis was done with gProfiler using *Taeniopygia guttata* as background species [95].

### ATAC-seq

**Data generation and processing.** We performed ATAC-seq for liver and spleen dissected from eight females (3 *corone*, 5 *cornix*) and seven males (4 *corone*, 3 *cornix*). For spleen, sample D_Ko-29 did not have enough material for ATAC-seq processing. For each biological replicate, two technical replicates were generated. We followed an ATAC-seq protocol specifically developed for frozen tissue [96]. In brief, ~ 10 mg of cryopreserved liver and spleen tissue samples were ground using a TissueRuptor (Qiagen, Hilden, Germany) in liquid nitrogen and subsequently homogenized in HB Buffer (see buffer recipes in Corces et al. 2017). Cells were counted with a hemocytometer and 50,000 cells were used for transposition. Transposition was carried out for 60 min in a 50μl reaction with 2.5 μl (100nM final) of Tn5 transposase (Illumina Nextera) per sample. Libraries were amplified using the primers published in Buenrostro et al. [54]. The number of necessary library PCR amplification cycles was determined by qRT-PCR using SYBR Green 10x in DMSO (Sigma-Aldrich) and NEBNext Master Mix 2x (New England BioLabs). DNA concentration of the resulting library was determined with a Qubit 2.0 Fluorometer (Life Technologies, Grand Island, NY) and its quality was assessed using an Agilent Bioanalyzer 2100 (Agilent Technologies, Santa Clara, CA). Libraries that passed the quality controls (bioanalyzer fragment distribution following nucleosomal patterning) were selected for sequencing. Two technical replicates were prepared per sample and sequenced (50 bp paired-end) on an Illumina HiSeq1500 sequencer in an inhouse sequencing facility at LMU Munich and on an Illumina NovaSeq S1 flowcell by the SNP&SEQ Technology Platform in Uppsala.

Sequencing data quality was assessed with FastQC (v0.11.5), and base pairs with a sequencing Phred quality score < 20 were removed. Nextera adapters were trimmed, and reads with a minimum length of 36 bp were selected for further mapping using Cutadapt (v1.9.1) [97]. Selected reads were mapped to the crow genome version 5.6. (NCBI accession number: GCA_000738735.4) using Bowtie2 (v2.3.2) [98], with the following parameters: bowtie2 –very-sensitive -X 2000, allowing for fragments of up to 2 kb to align. Only properly paired and uniquely mapped reads with a mapping quality score > Q20 were filtered out using samtools (v1.4) [99]. PCR duplicate reads were removed with Picard tools (v2.0.1).

**Quality control.** Following guidelines of ATAC-seq quality control assessment, we included the following indicators to judge data quality: (1) *Insert size distribution* following the expected nucleosomal spacing periodicity (**S20 Fig**) (2) *Spearman correlation coefficients* between technical replicates (**S21 Fig**) (3) The *number of usable reads* (4) The *fraction of reads in peaks* (FriP) as calculated with deepTools -plotEnrichment (**S22 Fig**) and (5) *ATAC-seq coverage enrichmentaroundt transcription start sites* (TSS), (**S23 Fig**). More details on quality control are provided in the **S2 Text**. Technical replicates that survived the quality check were merged. Biological replicates were not merged but were used as true replicates, ending up with 15 for liver and 14 replicates for spleen, which were then used for further peak calling. A summary of the usable reads can be found in **S10 Table**.

**ATAC-seq peak calling.** Differences in read coverage among samples can influence the sensitivity and specificity of peak identification. To avoid this, we down-sampled all the libraries after mapping and filtering to the read number of the smallest library. This resulted in ~60 million workable reads in liver and ~90 million in spleen for all samples (**S10 Table**). Even though we had a third more reads for spleen, we did not detect more peaks in this tissue, corroborating the fact that we reached a saturation peak-detection point. Down-sampling was

done with samtools view -s and coverage equality across libraries was checked by calculating the coverage for windows of 10 bp using deepTools -bamCoverage -bs 10 -p max (v2.27.1) [100]. No further normalization was done to the samples. Since sequence coverage (and hence the background considered by MACS2) differs between chromosomes due to large differences in GC content between micro- and macrochromosomes, peaks were called separately per chromosome using MACS2 (v2.1.0) macs callpeak -t -n -g 1e9 -f BAMPE [101]. Further functional annotation was performed with Homer (v1.0) including categories of transcription start and end sites, coding sequence, introns and intergenic regions [102].

**ATAC-seq analysis.** Commonly, two measures are used to assess broad-scale and localized chromatin environment: the number of ATAC-seq signals around genes (peak density) and their amplitude (peak height) [103]. In the context of dosage compensation and balance, interpretation of peak density is, however, challenged by ploidy differences. On the female Z chromosome, peak density reflects spacing of open chromatin regions of a single chromosomal copy. This allows direct comparisons of peak density to transcript levels. In males, however, the density measure integrates peaks from two chromosomal copies. A density value of, for instance, ten peaks surrounding a gene may then either reflect ten, fully overlapping peaks or five non-overlapping peaks on each of the two chromosomes. If open chromatin regions do not fully overlap, the density measure can be biased upwards in males and can no longer be directly related to transcript levels, nor to density values in females. Therefore, to score chromatin accessibility we restricted our analyses to peak height as a metric.

Peak height was calculated at gene level including the gene body and 20k - up- and downstream of the expressed gene, and at the regulatory level restricted to 1kb upstream of the transcription start site (TSS) or 1kb downstream of the transcription termination site (TTS) of the expressed gene (as defined by HOMER), excluding the gene body. Peak height was defined as the fold-change value estimated with MACS2 v2.2.7 [101]. We only worked with peaks with a significant support q-value $< 0.05$. Additionally, we also quantified peak height as raw counts in peaks and analyzed f:m differences in the Z using edgeR v3.15 [104]. For this, we recovered counts in peaks using the down-sampled bam files with featureCounts. To retrieve the raw counts for each biological replicate we generated a reference gtf file with consensus peaks across samples by selecting peaks shared in 80% of the samples and merging peaks with an overlap of 50%, for each tissue and sex separately. We then recovered raw read counts falling in these peaks in all of the biological replicates. Peaks were filtered out when cpm $> 1$ in 7 of the samples for both tissues and a generalized linear model was fitted as implemented in EdgeR.

Empirical estimates of mean peak height ratios of genes residing on the female vs. male Z chromosome were calculated as $Z_f{:}ZZ_m = median(Gene_i$ peak height$_{female}$)/ median(Gene$_i$ peak height$_{male}$). Z-chromosome to autosomes ratios (Z:A) for each sex were calculated as median(Z chromosome peak height)/median(autosomes peak height). We observed considerable variation in Z:A ratios across autosomes (**S4 Fig**). This variation can partly be explained by the differences in chromosome sizes present in the European crow and in birds in general [63]. Since there is a strong heterogeneity across chromosomes, we included chromosome and gene length as co-variables in the linear mixed models where a significant interaction with peak height was recovered. Throughout the manuscript model-based estimates are reported.

## 5mC methylation

**Data generation and processing.** We generated whole-genome bisulfite sequencing (WGBS) libraries for two female and two male individuals for both liver and spleen tissue. Post-bisulfite converted libraries were generated using the Illumina TruSeq Methylation Kit

(EGMK91324) following the manufacturer's protocol at the SNP&SEQ Technology Platform in Uppsala, Sweden. Raw reads were trimmed using Trim_Galore! v0.6.6 [105], clipping the first 9 bases from the 5' ends, and the last base of the 3' end, following standard post-bisulfite converted WGBS library protocols and as indicated by methylation bias plots (**S13 Fig**). Base and tile quality was assessed with FastQC v0.11.9 after trimming [88]. Cleaned reads were then aligned to the reference genome version 5.6 (NCBI: GCA_000738735.6 with default settings, followed by read de-duplication and extraction of CpG methylation state using Bismark v0.23.0 [106] (**S11 Table**). We retained only CpGs on assigned chromosomes (1–28 and Z) and removed any positions overlapping transition mutations (C–T, or G–A SNPs), identified from population resequencing data generated from a previous study (discussed below; [82]. We further excluded any sites overlapping repeats identified with RepeatMasker v4.1.1 [107], using the chicken repeat library and excluding short simple repeats (-nolow). We also created a CpG island track for assembly version 5.6 because of our interest in regulatory regions, using makeCGI [108], requiring stringent thresholds of a length of at least 250bp, a GC content > 60%, and an observed to expected CG ratio > 75%, resulting in 18,526 islands (**S12 Table**).

**Population resequencing reanalysis.** We reanalyzed population resequencing data for 15 male hooded (*C. (corone) cornix*) and 15 male carrion (*C. (corone) corone*) crows from an existing study [82] to identify transition SNPs which could confound WGBS methylation calls (*e.g.* a TG genotype at a CG site will incorrectly be called as non-methylated, it is therefore best practices to exclude C–T and A–G sites [109]). In short, resequencing data was trimmed with Bbtools v38.18 [110] and aligned to the crow reference genome using BWA v0.7.17 [111]. Read duplicates were removed with GATK v4.1.4.1 [112] and variants were called with Free-bayes v1.3.2 [113] without population priors. SNPs were filtered with bcftools [114], excluding INDELs and sites exhibiting the strongest allelic balance (AB), depth (DP), strand bias (SRP, SAP) and read placement (EPP) biases, identified from the top and bottom 2.5% distribution outliers of sites for each metric (only the bottom 2.5% were removed for SRP, SAP, and EPP because only low scores indicate bias). We furthermore removed sites with a quality (QUAL) below 20, a genotype quality (GQ) below 20, and required at least 75% of individuals to share a site with at least a genotype depth of 3 reads. We then retained only C–T or A–G SNP sites using bash.

**Genome-wide data analyses.** DNA methylation analysis was divided into gene-based analyses and CpG island-based analyses to determine the relative impacts of putatively increasing regulatory importance. We calculated methylation proportions (count methylated by cumulative read count) for each feature (*e.g.*, gene or CpG island) considering individual 5mC calls only if covered by at least 10 reads. Only features with scorable data within tissue-specific libraries were retained. Median and 95% confidence interval of sex differences (log2(f:m)) for each feature were drawn from 10,000 bootstrap sampling events. Sex-specific differentially methylated regions were called using a traditional differentially methylated point analysis using a beta-binomial regression on methylation read counts over the entire feature, implemented in DSS [115] within tissue libraries with sex as a covariate. To assess sensitivity to coverage requirements over the Z chromosome, we repeated bootstrap sampling and differentially methylated feature analyses with female-specific coverage thresholds over the Z (minimum 5 reads) and arrived at the same conclusions (**S14 Fig**). We calculated correlations between differences in gene expression (log2(f:m)) and methylation (log2(f:m)) using expression values directly for gene body methylation, and using the closest positional gene values for CpG islands, identified using bedtools v2.30 [116]. Statistical methylation differences between dosage balanced and unbalanced genes was analyzed using a Wilcoxon test.

**Identification of a male hypermethylated region.** Two male hypermethylated (MHM) regions have been identified in Galloanserae on the Z chromosome, although only one region has been suggested in other avian clades [44]. Here, we explored the possibility of an MHM region in the European crow by examining base-pair resolution DNA methylation divergence. We started our search by blasting the known MHM sequences [44] against the hooded crow reference. We further hypothesized that if the European crow harbored an MHM, it would exhibit male-specific hypermethylation and be significantly enriched in differentially methylated positions (DMPs) for sex-effects. To test this, we used the same filtered 5mC calls as above, but instead kept each individual base-pair position as a feature, and retained only sites shared across all available libraries ($n = 8$). Similarly, we retained calls only if covered by more than 10 reads, except for female libraries in which we allowed a relaxed 5 reads only along the Z chromosome. We then detected DMPs with a beta-binomial regression using tissue and sex as explanatory covariates, implemented in *DSS* using R [115]. Additionally, we calculated median divergence and 95% confidence intervals (both (f-m) and log2(f:m)) drawn from 10,000 bootstrap replicates for each site using R [117]. We then visualized divergence ((f-m) and log2(f:m)) and DMPs with karyoploteR [118].

## Identification of the pseudo autosomal region (PAR)

We took four independent approaches to identify the PAR in the European crow (see also **S1 Text**).

1. *DNA sequencing coverage*. Genomic regions on the Z belonging to the PAR are expected to have f:m$_{coverage}$ of approximately 1 for whole-genome shotgun sequencing data. For genomic regions outside the PAR we expect f:m$_{coverage}$ to be around 0.5. To estimate sex-specific coverage along the Z chromosome we used 150bp paired-end Illumina sequencing data from one female and one male crow with NCBI bioproject_id: PRJNA192205 [82]. Reads were mapped with BWA mem (v0.7.17) [111] to version 5.6 of the crow genome (NCBI: ASM73873v4). Only concordant paired reads with mapping quality >20 were kept using samtools view -u -h -q 20 -f 0x2 [99]. Read duplicates were removed with Picard tools and repeats were masked and filtered using RepeatMasker and the chicken repeat library (v4.0.7) [107]. Coverage was calculated with bamcoverage deepTools (v2.5.1) [100] in 50kb windows for the Z chromosome and chromosome 1 and 2 which are similar in size and GC content: bamCoverage—binSize = 50000—normalizeUsingRPKM -p 2. Differences in chromosomal coverage were assessed as follows: the coverage value of each 50 kb window along chromosomes 1,2 and Z was divided by the median coverage of these autosomes. This was done separately for the female and male sample. The normalized coverage was used to calculate log$_2$(f:m$_{coverage}$). Additionally, we assessed f:m$_{coverage}$ in the crow's Z in regions corresponding to gametologs. We used the set of gametologs identified in neoaves and in the flycatchers as published in [119]. We identified these sets of gametologs in the crow's Z by matching the annotation identifier in [119] to the identifiers in the crow's Z genome version 2.5. These regions were then subtracted in fasta format and mapped to the crow's genome v5.6 using lastZ (v1.04.00) [120]. We also checked on f:m$_{coverage}$ within each expressed gene on the Z in liver and spleen with the purpose of correlating ploidy and gene expression.

2. *Heterozygosity levels*. In heterogametic females, we expect to find heterozygous sites in the PAR as opposed to regions outside the PAR, where no heterozygosity should be found. We used GoldenGate genotype data from the Z chromosome (N = 129 SNPs) collected on 522 hooded and carrion crows and their hybrids (data described in Knief et al. [81]). For each

SNP, we calculated the observed heterozygosity in females (N = 277 individuals) and males (N = 245 individuals). We transferred SNP positions to crow genome version 5.6 and observed 15 heterozygous SNPs in females on the Z chromosome between 27,730 and 687,785 bp. To identify the gametologs in the crow's Z, we matched the Z annotation names to the list of gametologs identified in neoaves and the flycatches as published in [119]. The coordinates on the crow's Z corresponding to the two sets of gametologs, were used to retrieve f:m coverage values.

3. *Orthology*. In addition to the above approaches, we assessed synteny with the PAR of collared flycatcher (*Ficedula albicollis*; [121]). We aligned the Z of the flycatcher genome version 1.5 [122] using the genome aligner lastZ (v1.04.00) [120] and selected only those contigs with > 90% identity. Anchors to homologous regions between the species were plotted in R [117].

4. *Genome assembly*. The current assembly of the crow W chromosome has been generated by subtractive mapping to the high-quality assembly of a male individual including the PAR of the Z [84]. Alignment of the W to the Z chromosome should thus only include hits to the non-PAR region and none to the PAR region. Alignment of the W chromosome was conducted to the crow's genome v5.6 (NCBI: ASM73873v4) with lastZ (v1.04.00) [120]. The alignment was visualized with Circos v0.69–8 [123], provided they were longer than 1 kb with a percent identity above 70%. This process was repeated for the New Caledonian crow genome (RefSeq: GCF_009650955.1).

## Statistical analysis

Open chromatin states and 5mC methylation levels are expected to covary with a set of genomic features such as chromosome length and gene length, known to differ widely among avian chromosomes [63]. Therefore, categorical comparisons between the long Z chromosome (75 Mb) and all autosomes including micro-chromosomes (4.88–154.33 Mb; RefSeq GCF_000738735.5) are only meaningful if these factors are statistically controlled for. Accordingly, subsequent analyses focusing on differences between autosomes and the Z chromosome included these features as constitutive covariates where appropriate. Our measures of interest are also expected to vary in different parts of the chromosome (e.g. genic, intergenic). We therefore considered two scales of integration: actively expressed genes with their regulatory regions (gene-centred) and flanking regulatory regions without the gene body (regulatory). Statistical results for all measures are summarized across these levels in **S2 Table,** which also summarizes all f:m and Z:A ratios ($AA_f:AA_m$, $Z_f:ZZ_m$, $Z_f:AA_f$, $ZZ_m:AA_m$), both as raw estimates and parameter estimates with confidence intervals controlling for covariates. For chromosome comparisons, the Z chromosome was considered as a whole including the pseudoautosomal region (PAR). Effects of the PAR were then isolated in a second step.

**Linear models.** For the expression analyses, we included genes that were expressed in all individuals (zFPKM > -3). To fit the statistical models including ATAC-seq data, we constructed a dataset of orthologous peaks, in which we defined orthologous peaks as those overlapping for at least 50% of their lengths and that were present in at least two samples (singleton peaks were eliminated as they may represent errors). We included only orthologous ATAC-seq peaks associated with genes that were expressed in all individuals. In order to reduce the effect of ploidy, we conducted the following analyses: for peak height, we included only those orthologous peaks that were called in all individuals. We assume that this procedure reduces the effect of ploidy because a peak present in all individuals is also likely to be present on both copies of the male Z-chromosome. We fitted linear mixed-effects models with a Gaussian

error structure using the lme4 (v1.1–27.1) and lmerTest (3.1–3) R packages [124,125] for analyzing gene expression, ATAC peak height and 5mC methylation. We transformed the data to match normality (dependent variables: FPKM and ATAC peak height were $log_2$-transformed and percent 5mC was arcsine-square-root-transformed). It has been shown that Gaussian models are robust and that violating the normality assumption has only minor effects on parameter estimates and P-values [126]. For each dependent variable, we fitted whether the gene/ATAC peak/methylation window was located on an autosome or the Z chromosome (factor with two levels) and the sex of the individual (factor with two levels) as interacting fixed effects. We controlled for gene length and chromosome length where appropriate and Z-transformed these covariates. Individual ID and gene ID nested in chromosome ID were modeled as random effects. Exact model specifications are provided in **S2 Table.** We used the effects (v4.2–0) R package [127] to obtain the four parameter estimates (autosomes in females, autosomes in males, Z in females, Z in males). We calculated ratios on these parameter estimates and bootstrapped 95% confidence intervals using the bootMer() function in lme4 with 1,000 replicates. These ratios are reported in **S2 Table** together with model estimates. Fixed effect estimates were considered significant when $P < 0.05$. The variance explained by the random effects was estimated using the rptR package (v0.9.22) [128]. To explore the relationship between peak height and gene expression we fitted the following linear mixed model: peak_-fold_change ~ $log_2$(gene_fpkm) + tissue + (1 |geneID) + (1 | sampleID) + (1|chrID), working only with the highest peak linked to an expressed gene. We did this analysis in different expression classes defined by their quartile distribution.

In a second set of models, we restricted the data to the Z chromosome and separated the PAR from the rest of the Z (nonPAR), such that we estimated four parameters (nonPAR in females, nonPAR in males, PAR in females, PAR in males). We controlled for gene length (Z-transformed) by fitting it as a covariate and included gene ID as random effect. When analyzing peak height, we also included individual ID as a second crossed random effect. We used the effects (v4.2–0) R package (Fox & Weisberg 2019) to obtain the four parameter estimates, calculated ratios on these parameter estimates and bootstrapped 95% confidence intervals using the bootMer() function in lme4 with 1,000 replicates. These ratios are reported in **S2 Table**.

**Identification of female to male ratios along the Z-chromosome.** To identify putative clusters of dosage compensated genes on the Z chromosome and to identify regions on the Z enriched or depleted for peak height, we performed a sliding window analysis. We used a window size of 688 kb (the size of the PAR) and 1 Mb with a sliding step of 344 kb (half PAR size) and 100 kb respectively. Gene expression and ATAC-seq peak metrics on the PAR were used as reference to identify deviating regions along the Z chromosome. A Fisher's exact test was performed for each window to detect enrichment or depletion of dosage compensated genes along the Z (categories as defined above), with a Bonferroni correction for multiple testing. To detect regions on the Z enriched or depleted in number of peaks, we additionally used a Wilcoxon test as alternative = "less" and alternative = "greater". To further survey the Z for potential clustered centers of regulation, we conducted an autocorrelation analysis using the Acf() function from the R package forecast v5.8 [129]. In order to identify peaks with significantly different heights between females and males on the Z, we used the gtf file with consensus-peak coordinates we first identified consensus peaks present in either of the sexes. We selected peaks present in 80% of both female and male samples. Peaks overlapping by more than 50% between individuals were merged into a single peak to avoid peak redundancy across samples. For these consensus-peak coordinates we used extracted peak heights as peak_fold_change computed in MACS2.

Only peaks were considered, where at least three individuals had non-zero values. Differences in the peak height distribution between females and males for a given window was tested in R using a Wilcoxon test. Raw read counts were retrieved for the consensus-peaks using FeatureCounts [130] and an EdgeR [104] analysis of the Z chromosome was performed, normalizing libraries having 1 count per million in at least seven libraries.

## Supporting information

**S1 Fig. Raw Z:A ratio for gene expression, shown for each autosome in females (light blue) and males (dark blue).** Confidence intervals (95%) drawn from 10,000 bootstraps. Vertical dashed lines show the values for no dosage compensation (0.5) or full dosage compensation (1).
(TIFF)

**S2 Fig. Autocorrelation of gene expression present on the Z chromosome.** Red vertical line shows the PAR border. Blue lines represent autocorrelation values significantly different from zero. Except for the PAR region, there is no clear trend of clusters that would point at local centers of dosage compensation. Liver: upper panel. Spleen: lower panel.
(TIFF)

**S3 Fig. Boxplot showing the distribution of the number of associated peaks per expressed gene in the autosomes and the Z chromosome in a region of 20 kb upstream and downstream of the focal gene.**
(TIFF)

**S4 Fig. Z:A ratio for peak height shown for each autosome in females (light blue) and males (dark blue).** Confidence intervals (95%) drawn from 10,000 bootstraps. (A) shows gene-centered Z:A ratios. (B) shows Z:A only in up- and down-stream regions of expressed genes excluding the gene body.
(TIFF)

**S5 Fig. Boxplots presenting $log_2$ gene-centred peak height distribution broken down into four quantiles (Q1 –Q4).** A: autosomes, Z: Z chromosome. Upper panel liver, bottom panel spleen.
(TIFF)

**S6 Fig. Boxplots showing gene-centered peak height distribution including different size regions around the gene body (1 kb, 10 kb,15 kb and 20 kb) in females (F) and males (M) on the Z chromosome.**
(TIFF)

**S7 Fig. Boxplots showing sequencing coverage distribution.** Coverage was calculated at intervals of 100kb. Liver: AAf:AAm = 0.90, Zf:ZZm = 0.76, Zf:AAf = 0.70, ZZm:AAm = 0.84. Spleen: AAf:AAm = 1.02, Zf:ZZm = 0.92, Zf:AAf = 0.73, ZZm:AAm = 0.81.
(TIFF)

**S8 Fig. Frequency distributions of % 5mC CpG DNA methylation underlying data in S2C Table for the CpG island and gene-centric analytical subdivisions for the two biological replicates within each tissue.**
(TIFF)

**S9 Fig. CpG methylation levels surrounding ATAC-seq features.** For each sample (n = 8), median methylation levels were calculated in the +/- 10-KB region surrounding a sample's specific ATAC-seq peak. This 20-kb flanking region was divided into 100 running windows of 200-bp, and the median percent methylation value of each tile was assigned. Before averaging,

peaks were divided into more-expressed or less-expressed delimitations based on the log10 (FPKM) values of their associated genes, delineated by either peaks above or below the 30% interquartile range of gene expression values, calculated independently for autosomal and Z chromosome genes at the sample-specific level.
(TIFF)

**S10 Fig. Correlations between sex-specific DNA methylation divergence.** (log2(F/M)) and expression (log2(F/M)) across both tissues for both CpG islands and directly over gene bodies for Z-linked transcripts. CpG island expression level was determined as the expression level of the closest genomic transcript. Spearman's rho is written for each facet, and for each dosage compensation state (dosage balanced, or unbalanced, as identified from expression data).
(TIFF)

**S11 Fig. Sex-specific CpG methylation divergence on the Z chromosome.** Differentially methylated features (genes or CpG islands) were identified with a beta-binomial regression using methylation counts implemented with a general experimental design framework in DSS. FDR-corrected -log10(p-values) for sex divergence are plotted along the bottom first and third panels for liver and spleen, respectively, with the dashed line indicating FDR-corrected significance (0.05). Bootstrap sampling log2(F/M) values are shown along bottom second and fourth panels.
(TIFF)

**S12 Fig. Sex-specific CpG methylation divergence genome-wide.** Differentially methylated features (genes or CpG islands) were identified with a beta-binomial regression using methylation counts implemented with a general experimental design framework in DSS. FDR-corrected -log10(p-values) for sex divergence are plotted along the bottom first and third panels for liver and spleen, respectively, with the dashed line indicating FDR-corrected significance (0.05). Bootstrap sampling log2(F/M) values are shown along bottom second and fourth panels.
(TIFF)

**S13 Fig. Quality control on the 8 WGBS libraries, providing two male and two female replicates for both liver and spleen.** Panel A indicates methylation levels averaged across all reads in a library along read position, used to identify systematic methylation biases along read lengths. Sample name: Individual: S_Up_H59/H60; tissue:liver (L) or spleen (M); age: adult (ADL); sex: male (M) or females (F)). Panel B indicates total 5mC positions called for each sample, as well as sites retained after filtering for C-T and G-A transition SNPs, and retaining sites on assembled chromosomes.
(TIFF)

**S14 Fig. Sensitivity analysis showing two metrics of 5mC methylation sex divergence (f-m and log2(f:m)) across the Z chromosome for three feature-based approaches (cgi: CpG islands, chromosome: entire chromosome, and gene: gene-based).** Features were classified as female- or male-biased if they were significantly different for sex from a beta-binomial regression on read counts, and showed either a positive log2(f/m) or f-m greater than 25% (female-biased), or a negative log2(f:m) and f-m less than -25% (male-biased). This analysis also retained female-specific 5mC calls on the Z chromosome which contained 5 reads, thereby providing a lower coverage threshold for the single-copy female chromosome. No differences between these results and the primary manuscript indicate sensitivity issues with technical filtering or analytical parameters.
(TIFF)

**S15 Fig. Boxplot of female to male read coverage distribution from whole genome shotgun sequencing of one individual of each sex as calculated in 50 kb windows, in chromosome 1 (chr1), chromosome 2 (chr2), the pseudoautosomal region (PAR) and the Z chromosome without the PAR.** F/M coverage ratio on the shown autosomes and the PAR lie near a ratio of 1 (red line), whereas the hemizygous part of the Z lies close to a ratio of 0.5 (grey line).
(TIFF)

**S16 Fig. Boxplots showing the distribution of f:m DNA sequencing coverage on regions of the European crow's Z chromosome that map to the identified gametologs in Neoaves (Bellott et al., 2017 [119]).** Red horizontal lines mark f:m = 1 and f:m = 0.5. Upper panel: data shown for liver. Lower panel: data shown for spleen.
(TIFF)

**S17 Fig. Boxplots showing the distribution of f:m DNA sequencing coverage on regions of the European crow's Z chromosome that map to the identified gametologs in flycatcher (Bellott et al., 2017 [119]).** Red horizontal lines mark f:m = 1 and f:m = 0.5. Upper panel: data shown for liver. Lower panel: data shown for spleen.
(TIFF)

**S18 Fig. Female:male DNA sequencing coverage ratio for every expressed gene of the Z chromosome in liver and spleen.** Each boxplot represents the f:m coverage ratio within the coordinates of each gene. Boxplot colors represent three dosage compensated states: compensated, partially compensated and not compensated. Solid horizontal line marks f:m = 1 and dashed horizontal line marks f:m = 0.5. Red vertical dashed line shows the PAR limit.
(TIFF)

**S19 Fig. Chromosome Z to W alignment.** Contigs in blue represent the W chromosome and in gray the Z. The PAR in the Z chromosome is indicated in orange. Purple lines indicate matching regions between the W and the Z chromosome.
(TIFF)

**S20 Fig.** ATAC-seq insert size distribution for all biological replicates for spleen (left) and liver (right). The position of the first nucleosome can be observed between ~150–250 bp.
(TIFF)

**S21 Fig. Across-sample correlation between technical and biological replicates using genome coverage as the correlation parameter in liver and spleen.**
(TIFF)

**S22 Fig. Percentage of reads mapped in peaks for pooled samples in liver and spleen.**
(TIFF)

**S23 Fig.** Upper panels: summary of coverage density +/- 2kb of the TSS, in spleen (left) and liver (right). Bottom panels: Enrichment of mapped reads presented as heatmaps along the defined TSS, in spleen (left) and liver (liver).
(TIFF)

**S1 Text. Identification of the pseudo-autosomal region (PAR).**
(DOCX)

**S2 Text. Pre-processing of ATAC-seq data.**
(DOCX)

**S1 Table. Annotated genes residing on the Pseudo Autosomal Region (PAR).**
(XLSX)

**S2 Table. Description of statistical models and model outputs.** Genome-wide statistical models were of the form lmer(Y ~ AZS + (1|indID) + (1|chrID/xxxxID)), with Y being (A) log2(FPKM)-values, (B) log2(Fold change)peakheight and (C) asin(sqrt(5mCG)). AZS is a factor with four levels: Female Autosomes (AAf), Male Autosomes (AAm), Female Z (Zf) and Male Z (Zm). We included the identity of the sample (indID) as a random effect. Depending on the unit of measurement (i.e. genes, ATAC peaks, 5mC sites), we included either gene (geneID), peak (peakID) or 5mC site (siteID) nested in chromosome (chrID) as additional random effects. In (A) we included chromosome and gene lengths as covariates and in (B) and (C) we included chromosome length as a covariate. Models were fitted for genes (including transcription start site, exons, introns and transcription termination sites) and regulatory regions (transcription start and transcription termination sites) and for liver and spleen separately. For the gene-centered analyses, we included log(gene length) as an offset term and in the analyses of the regulatory regions we included Z-transformed gene length. We did not observe overdispersion. We also fitted models focusing on chrZ, separating the PAR from the rest of the chromosome as lmer(Y ~ AZP + (1|indID) + (1|xxxxID)) or glmer(N Peaks ~ AZP + (1|peakID)). Here, AZP is a factor with four levels: Female PAR (PARZf), Male PAR (PARZZm), Female rest of chromosome Z (Zf) and Male rest of chromosome Z (ZZm). The random effect structure was similar to the above models, except that we left out chrID as a random effect. In (A), (B) and (C) we provide sample sizes and the variance explained by each random effect (Vxxx), parameter estimates with their standard errors, z- and p-values and also the ratios between parameter estimates with bootstrapped 95% confidence intervals.
(XLSX)

**S3 Table. Enrichment analysis of dosage compensated genes along the Z.**
(XLSX)

**S4 Table. Gene ontology enrichment analysis (GO) of shared expressed balanced genes on the Z chromosome.**
(XLSX)

**S5 Table. Peak height enrichment analysis along the Z chromosome.**
(XLSX)

**S6 Table. F:m peak ratios.** Pvalues correspond to Wilcoxon test comparing peak heights between females and males. NA correspond to those peaks where the peak number was less than three.
(XLSX)

**S7 Table. EdgeR analysis on the Z assessing differences in chromatin accessibility between females and males in A) liver and B) spleen.**
(XLSX)

**S8 Table. Mean, median, and 95% confidence intervals for F/M methylation values in dosage balanced and unbalanced genes and CpG islands.**
(XLSX)

**S9 Table. SRA identifiers for the sequenced data.**
(XLSX)

**S10 Table. Assessment of usable reads for each ATAC-seq library sequenced after quality filtering.**
(XLSX)

**S11 Table. Mapping efficiency, CpG calls, bisulfite conversion rate, and SRA accessions for whole-genome bisulfite sequencing libraries (WGBS).**
(XLSX)

**S12 Table. Coordinates and features of the 18,526 CpG islands, identified with makeCGI.** A threshold of > 250 BP, > 60% GC content, and an Observed/Expected ratio > 0.75 was required to delineate an island.
(XLSX)

**S13 Table. Position on the Z chromosome of the European crow genome v5.6 corresponding to the gameologs identified in neoaves and flycatcer (Bellot et al. 2017 [119]).**
(XLSX)

**S14 Table. Number of mapped reads after filtering steps described in the methods section.** The libaries were down-sampled to the library with the lowest number of reads. The normalization factor indicates whichi proportion of original reads was kept. Data is shown seperately for liver and spleen.
(XLSX)

## Acknowledgments

We express our gratitude to Sven Jakobsson for providing the infrastructure for animal husbandry at Tovetorp research station. We would also like to thank Christen Bossu, Jelmer Poelstra and Matthias Weissensteiner for their contribution in obtaining samples. Kristaps Solokovskis, Thomas Giegold, Nils Andbjer, Tamara Volkmer, Barbara Martinschitsch and Luisa Sontheimer under guidance of Julia Buskas provided invaluable support in raising and maintaining the captive crow population. Martin Wikelski, Inge Müller and additional staff from the Max-Planck-Institute for Ornithology in Radolfzell facilitated sampling in Germany and transport to Sweden. We are further grateful to Gabriele Kumpfmüller with whom the ATAC-seq protocol was established. The Swedish sequencing facility is part of the National Genomics Infrastructure (NGI) Sweden and Science for Life Laboratory. We further thank Stefan Krebs and Helmut Blum at the Gene Center Munich for discussions and sequencing the ATAC-seq libraries.

## Author Contributions

**Conceptualization:** Ana Catalán, Jochen B. W. Wolf.

**Data curation:** Ana Catalán, Justin Merondun.

**Formal analysis:** Ana Catalán, Justin Merondun, Ulrich Knief.

**Funding acquisition:** Jochen B. W. Wolf.

**Investigation:** Ana Catalán.

**Methodology:** Ana Catalán, Justin Merondun.

**Supervision:** Jochen B. W. Wolf.

**Validation:** Ana Catalán, Justin Merondun.

**Visualization:** Ana Catalán, Justin Merondun.

**Writing – original draft:** Ana Catalán.

**Writing – review & editing:** Ana Catalán, Justin Merondun, Ulrich Knief, Jochen B. W. Wolf.

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
