## [Decision Letter · Decision Letter 0]

2 May 2023

Dear Dr Catalán,

Thank you very much for submitting your Research Article entitled 'Epigenetic mechanisms of partial dosage compensation in an avian, female heterogametic system' to PLOS Genetics.

The manuscript was fully evaluated at the editorial level and by independent peer reviewers. The reviewers appreciated the attention to an important topic but identified some concerns that we ask you address in a revised manuscript.

We therefore ask you to modify the manuscript according to the review recommendations. Your revisions should address the specific points made by each reviewer. Please also ensure that your genomic data are made available before resubmitting your revised manuscript.

Yours sincerely,

Marnie E. Blewitt

Academic Editor

PLOS Genetics

John Greally

Section Editor

PLOS Genetics

Reviewer's Responses to Questions

**Comments to the Authors:**

Reviewer #1: The paper shows the potential of combining gene expression data (RNA-Sequencing) and genome accessibility data (ATAC-Sequencing) in a non-model organism. In general, I think the experimental design and the analysis are of a very good standard. Even though it was already known that incomplete dosage compensation exists in crows, the paper nicely demonstrated how dosage compensation mechanisms reflects on chromatin level changes. Ultimately, however, the experimental design chosen here do not reveal the underlying molecular mechanism, but only a correlation between chromatin accessibility and gene expression. This the authors even admit themselves (e.g. in the text “We observed that gene expression covaried with peak height (Figure 4A) and observed that the highest peak is more often found in promoter regions than other parts of the gene or intergenic regions (Figure 4B).”) It is therefore important that abstract and text will be rephrased accordingly.

Otherwise, I think the paper makes a good case for publication in PLOS Genetics, if the points below are addressed. Lastly, I want to congratulate the authors on the methods paragraph, I think it has been really carefully written and contains many important details.

Specific comments:

• The authors suggest a chromosome-wide epigenetic mechanism on the female Z-chromosome in female crows. However, it was also shown that the dosage compensation/dosage balance is incomplete and the majority of genes are unaffected. For example, the authors write that “Using a sliding window analysis employing windows of different sizes (PAR equivalents of 688 kb or 1 Mb) no significant clustering of compensated genes was found …” – wouldn’t this rather speak for a gene-by-gene manner rather than a chromosome-wide upregulation? The authors should explain and adress how the observed upregulation in a gene-by-gene manner and a chromosome-wide epigenetic mechanism are compatible.

I would like to have a bit more information about the genes which are affected by the DC/DB mechanism, or even a little follow up on those.

• What kind of genes are affected? Are the Genes suspected to be dosage sensitive like for example genes that code for proteins that are subunits of bigger complexes? Maybe one could even do a GO analysis or compare with data from different species (from other studies).

• Are the affected genes overlapping in both tissues?

• Is there any correlation between status of compensation and expression level?

I think it is great that the study takes different tissues into account. The results show differences between the tissues (Spleen/Liver) in terms of DC/DB and 5mC-Methylation. I think these differences could be addressed in more depth.

• Are the same Genes affected (see 2.)?

• Could it be a concern that liver cells are often polyploid? (or is this not the case in birds?)

• Should we conclude from the results that future studies should look at more tissues to get a more complete picture of the mechanisms in the organism?

DNA methylation data

• Could the authors briefly explain in the text the state-of-knowledge about DNA methylation in birds? Does it occur at the same sites as in mammals? The analysis implies that CpG islands also exists in birds, but can this perhaps be shown? Is there any evidence that DNA methylation is repressive in birds?

• Since the Z chromosome is upregulated, I did not get the concept of why one would study a repressive mark per se? If the idea is that there is less methylation than in the ancestral state – is there a way that the authors could use DNA methylation data from an outgroup (chicken (?)) and ask whether the ancestral DNA methylation levels have been “erased” in crows? I don’t think this is a critical point, but perhaps (if the suggested analysis is not possible), the interpretations could be a bit toned-down.

Text

• “epigenetic control mediated by differential chromatin accessibility and to a lesser extent 5mC methylation between the sexes” in the abstract is an overstatement of the findings – if genes are more or less expressed they will obviously show differences in accessibility and/or 5mC, but this does not demonstrate by any means a causality for dosage compensation and/or balance, but instead is a correlation. This sentence from the abstract is just an example, but the entire text has to be adjusted accordingly.

• The authors should eliminate the word “epigenetic” from their text, because the analyses do not by any means address the inheritance patterns of the observed marks. Likely, all the observed effects are purely genetic. I guess the authors mean “chromatin-based regulation” and should use this phrasing instead.

• Intro: “For example, in mammals one of the X chromosomes is randomly targeted for inactivation in the homogametic sex by nucleosomal condensation” – not sure what is meant with “nucleosomal condensation” – please rephrase.

• Intro: “The active X chromosome is then hyper-transcribed to equalize expression levels to the (diploid) autosomal dose (Di and Disteche, 2006).” Rephrase - This statement is not correct as such, the upregulation of the active X in mammals involves multiple layers, including transcript stability and translational regulation and its mechanism is not full resolved yet.

• Intro: “In heterogametic ZW systems, with the exception of several moth species (Huylmans et al., 2017; Smith et al., 2014), …” also other ZW systems show full compensation, e.g. brine shrimp Artemia franciscana, turtle Apalone spinifera or Schistoma mansoni (at least in some tissues). Please rephrase.

• The discussion is a bit lengthy, perhaps one could shorten it a bit.

• Perhaps I missed it, but when did the Z chromosome of crows evolve? Is it the same Z as chicken (i.e. the chicken / crow Z evolved before the species split)? It would be helpful to add this info in the text.

Figures:

• Figure1A: f:m ratio for the PAR region is above the expected value of 1 (almost 1.5). As the hemizygous parts are also a bit a bit over 0.5, there is likely to be technical explanation.

• Figue1C: I am not quite sure if this graphic tells anything relevant for the understanding of the paper. It is also not mentioned in the text.

Reviewer #2: This manuscript takes a detailed look into the mechanism of partial dosage compensation in crows, and shows evidence for a chromosome wide epigenetic regulation mechanism. This has been previously hypothesised, but not tested so this manuscript nicely fills this gap in knowledge. The manuscript is well written and has sufficient data to test the hypotheses. I have very few comments.

The main one is that although the authors have produced and analysed a nice dataset the authors have not made the data nor their code available! – this should be done prior to submission following PLoS’ data policy.

Fig.2. Please use more contrasting colours – I find it hard to see the difference between the two blues.

Reviewer #3: Dosage compensation as a result of sex chromosome differentiation is achieved by different mechanisms and in the past has been predominately investigated at the transcriptional level (with the exception of human and mouse and to some extend chicken). The work is thorough, state of the art and does provide novel aspects on dosage compensation on the chromatin level that are important for this area of research. The data on DNA methylation and chromatin accessibility in relation to gene expression do provide a step forward in understanding how gene and chromosome organisation evolved during sex chromosome evolution and also provides important revision of the idea of region specific differential methylation in the ZZ/ZW system.

Overall I would encourage the authors to revise the manuscript so it is a bit easier for readers outside the specific area (including the title which I find a bit cryptic).

The authors could consider to just describe that sex chromosome differentiation and the associated gene loss from the heteromorphic sex chromosomes has led to various mechanisms to achieve dosage compensation. I personally find the framing around genetic sex determination a bit of a distraction this is fundamentally a result of sex chromosome differentiation.

Abstract: Why is it relevant that the crows were raised under garden conditions?

Line 99: “trapped in an arrested stage” what does this mean? Undifferentiated sex chromosomes have been described in several species

Lone 8-11, I find “the former” ,…, “the latter” always cumbersome and I suggest to define dosage balance and dosage compensation directly

124: “in some species” has a MHM been show for any other species that chicken? This becomes clearer in the discussion.

The authors could consider moving the description of the crow karyotype to the introduction.

Figure 1C shows a comparison between crow and flycatcher. There is no mention in the results section about what has been observed.

Table 1 : I suggest to shorten the description in the table header and describe much of it in the actual results text.

Line 373: “we found no orthologous region…” what does this mean? Did you look at the orthologous region of the chicken MHM? That should be spelled out.

Discussion: the discussion should be less technology heavy in the description of the results in my opinion. Avian dosage compensation has always been challenging to quantify and this paper does provide very sound data on 5mC and chromatin accessibility that shows no differential methylation (apart from spleen) but differential chromatin accessibility.

Line 414: “see Chen et al.? Why?

Lines: 425-440: although a bit of a side story I found these observations really interesting. Not sure how to better integrate in the manuscript.

Line 445: In the discussion I suggest to replace peak high by directly describing if chromatin was more accessible or not.

Line 447: the regions of differential accessibility are of interest but I feel they are not discussed very well, so these regions show a female bias in accessibility, what are the sequence features in this region, I take it that this is independent of 5mC but might be worth to spell out again.

Line 461: “Throughout the manuscript”?Again this section makes an important point but it seems a bit disconnected.

It is great to see the numbers of individuals and looking at more than one tissue. Is there anything noteworthy in terms of individual differences in the assays, also why liver and spleen and do the authors have any idea about the differences they observed in spleen?

In mammals the field has now moved to look at this from the proteom level and this may also be mentioned in this paper as a future direction in the avian clade.

Overall a really thorough study by the authors with important insights into Z chromosomes expression between sexes.

**Have all data underlying the figures and results presented in the manuscript been provided?**

Reviewer #1: Yes

Reviewer #2: Yes

Reviewer #3: Yes

PLOS authors have the option to publish the peer review history of their article (what does this mean?). If published, this will include your full peer review and any attached files.

Reviewer #1: No

Reviewer #2: No

Reviewer #3: No

---

## [Decision Letter · Decision Letter 1]

7 Aug 2023

Dear Dr Catalan,

We are pleased to inform you that your manuscript entitled "Chromatin accessibility, not 5mC methylation covaries with partial dosage compensation in crows" has been editorially accepted for publication in PLOS Genetics. Congratulations!

Yours sincerely,

Marnie E. Blewitt

Academic Editor

PLOS Genetics

John Greally

Section Editor

PLOS Genetics

Comments from the reviewers (if applicable):

Reviewer's Responses to Questions

**Comments to the Authors:**

Reviewer #1: I thank the authors for their careful revision. I recommend acceptance.

One small improvement suggestion for Figure 5: Perhaps I missed it but do panels A/B/C correspond to liver and D/E/F to spleen? I didn't find the information in the legend either. It would be great if this can be labelled clearly.

Reviewer #2: The authors have now provided their data and it is accessible, however they have not put a link to their code in the data accessibility statement (just in the response letter). Note that the link at the moment is a GitHub link - the authors should use a stable archive for code for the final version (as GitHub repos get deleted!), e.g. zenodo.

Otherwise, nice paper!

Reviewer #3: Thanks to the authors for thoroughly addressing the comments and suggestions. I offer some very minor comments on the revised ms:

Line 69: I suggest to rethink "appears" sex chromosome differentiation most certainly can involve Y or W degeneration (keeping in mind there are are also undifferentiated sex chromosomes that are evolutionarily stable).

Line 259: there seems something wrong with this sentence.

**Have all data underlying the figures and results presented in the manuscript been provided?**

Reviewer #1: Yes

Reviewer #2: Yes

Reviewer #3: Yes

PLOS authors have the option to publish the peer review history of their article (what does this mean?). If published, this will include your full peer review and any attached files.

Reviewer #1: No

Reviewer #2: No

Reviewer #3: No

**Data Deposition**

http://datadryad.org/submit?journalID=pgenetics&manu=PGENETICS-D-23-00282R1

**Press Queries**

---

## [Editor Report · Acceptance letter]

20 Sep 2023

PGENETICS-D-23-00282R1 

Chromatin accessibility, not 5mC methylation covaries with partial dosage compensation in crows 

Dear Dr Catalán, 

We are pleased to inform you that your manuscript entitled "Chromatin accessibility, not 5mC methylation covaries with partial dosage compensation in crows" has been formally accepted for publication in PLOS Genetics! Your manuscript is now with our production department and you will be notified of the publication date in due course.

With kind regards,

Judit Kozma

PLOS Genetics

On behalf of:
